# Evaluation of Probiotic Properties of *Pediococcus acidilactici* M76 Producing Functional Exopolysaccharides and Its Lactic Acid Fermentation of Black Raspberry Extract

**DOI:** 10.3390/microorganisms9071364

**Published:** 2021-06-23

**Authors:** Young-Ran Song, Chan-Mi Lee, Seon-Hye Lee, Sang-Ho Baik

**Affiliations:** Department of Food Science and Human Nutrition, Jeonbuk National University, Jeonju 561-756, Korea; gunandrang@hanmail.net (Y.-R.S.); 131536@naver.com (C.-M.L.); glgkglgk0202@naver.com (S.-H.L.)

**Keywords:** *Pediococcus acidilactici*, exopolysaccharide, lactic acid fermentation, black raspberry, antioxidant, flavors

## Abstract

This study aimed to determine the probiotic potential of *Pediococcus acidilactici* M76 (PA-M76) for lactic acid fermentation of black raspberry extract (BRE). PA-M76 showed outstanding probiotic properties with high tolerance in acidic GIT environments, broad antimicrobial activity, and high adhesion capability in the intestinal tract of *Caenorhabditis elegans*. PA-M76 treatment resulted in significant increases of pro-inflammatory cytokine mRNA expression in macrophages, indicating that PA-M76 elicits an effective immune response. When PA-M76 was used for lactic acid fermentation of BRE, an EPS yield of 1.62 g/L was obtained under optimal conditions. Lactic acid fermentation of BRE by PA-M76 did not significantly affect the total anthocyanin and flavonoid content, except for a significant increase in total polyphenol content compared to non-fermented BRE (NfBRE). However, fBRE exhibited increased DPPH radical scavenging activity, linoleic acid peroxidation inhibition rate, and ABTS scavenging activity of fBRE compared to NfBRE. Among the 28 compounds identified in the GC-MS analysis, esters were present as the major groups. The total concentration of volatile compounds was higher in fBRE than that in NfBRE. However, the undesirable flavor of terpenes decreased. PA-M76 might be useful for preparing functionally enhanced fermented beverages with a higher antioxidant activity of EPS and enhanced flavors.

## 1. Introduction

Although various exopolysaccharides (EPS) have been applied in the food industry as food additives for natural viscosifiers, emulsifiers, stabilizers, texturizers, and mouthfeel agents, they have also received great attention for human health because of their antitumor, anti-ulcer, antioxidant, blood glucose-regulating, and UV radiation-protecting activities [1]. Microorganism-derived EPS play an important role as technological substances in fermented products because their distinctive physical properties contribute to texture, taste perception, and stability in foods [2]. Different types of EPS derived from different microbial sources have different chemical structures and physiological functions. Lactic acid bacteria (LAB) are important for the production of valuable substances, such as mannitol, conjugated fatty acids, and EPS, during fermentation [3,4,5]. Particularly, EPS derived from LAB possess other functions, including medical, cosmetic, and pharmaceutical effects in human usage [6]. LAB or EPS have been certified as probiotics, but LAB with functional properties such as active EPS production have received little attention.

Many LAB strains considered as probiotics are effective against diverse gastrointestinal (GI) exertions by adhering to the gut mucosa and exerting beneficial effects on human health [7]. For use as probiotic bacterial strains, certain imperative characteristics are required to survive in acidic conditions and the presence of bile salts and to adhere to the intestinal tract [8]. Other beneficial effects, such as modulation of lactose intolerance, reduction of cholesterol levels, and regulation of the immune system, are also considered as additional criteria for the selection of probiotics [9]. Additionally, the physical properties of EPS enhance bacterial colonization of probiotic bacteria in the GIT, supporting their functional features [10]. LAB starter cultures with interesting functional characteristics and technological and probiotic properties have been isolated mainly from traditional fermented products [8].

Black raspberry (*Rubus coreanus Miquel*, BR), which is grown worldwide, is dark purple in color. Its natural colorant properties are used as a base for products such as ice-cream and sorbet. BR extract (BRE) can also be mixed with many beverages to enhance flavor and balance sweetness and tartness. However, the demand for BR has increased in other ways because of the expanded nutritional knowledge on anticancer, antibacterial, and antioxidant activities [11,12]. Due to the abundance of phenolic compounds, such as flavonoids (anthocyanins, flavonoids, and flavanols), tannins (proanthocyanins and ellagitannins), and phenolic acids (hydroxybenzoic and hydroxycinnamic acids), many products, such as sugar extract, wine, and vinegar, have been developed using BRE. Many studies have indicated that the chemical constituents change the raw BRE, and its fermentation might change the functional aspects, such as the antioxidant activity. Nevertheless, there is a lack of summarized data on the physiological aspects of different preparation approaches. To the best of our knowledge, this is the first study to use LAB for enhancing the functional features of BRE by expressing the functional component, EPS.

Previously, *Pediococcus acidilactici* M76 (PA-M76), which was isolated from *makgeolli* (traditional Korean rice wine) in our lab, produced a large amount of functional EPS in laboratory medium, showing high antioxidant and cytoprotective activity against alloxan-induced cytotoxicity. Moreover, it had a lipid-lowering effect in high-fat diet-induced obese mice [13]. The purified EPS from PA-M76 is a glucan consisting of glucose units with a molecular mass of approximately 67 kDa, which is unusually small compared to previously published figures [14]. Thus, our study aimed to evaluate the probiotic potential of EPS-producing PA-M76 and apply it to the preparation of functional healthy beverages of BRE. We determined the low pH and bile tolerances, intestinal adhesion capacity, and immune-stimulatory activity of PA-M76, as well as the optimal lactic acid fermentation conditions for optimized EPS production, antioxidant activity, and the profile of volatile compounds in the fermented BRE.

## 2. Materials and Methods

### 2.1. Bacterial Strain and Growth Conditions

PA-M76 isolated from Korean traditional rice wine, *makgeolli*, as a functional EPS producer by Song et al. (2013, patent no. KACC91683P) was routinely prepared in MRS broth (Difco Laboratories, Detroit, MI, USA) and grown at 37 °C for 18 h. Additionally, *P. pentosaceus* (ATCC 33316, PP), which is a widely studied probiotic as *Pediococcus* sp., and *L. rhamnosus* GG (ATCC 53103, LGG), a representative probiotic bacterium, were used as reference controls for the evaluation of the probiotic potential of PA-M76 [15].

### 2.2. Acid and Bile Tolerance

The in vitro survival ability of PA-M76 under conditions stimulating the GIT was determined using the method described by Oh and Jung (2015), with slight modifications. The active strains grown in MRS were centrifuged (5000× *g*, for 10 min at 4 °C), washed twice, and re-suspended in sterilized 50 mM phosphate-buffered saline (PBS) adjusted to pH 2.5 or 7.0. The cells (10^9^ colony forming units [CFU]/mL) were incubated at 37 °C for 3 h, and the resistance to low pH was assessed by measuring the survival of bacterial colony counts on MRS agar plates. Bile tolerance was determined by incubating LAB cells in MRS broth containing 0.3, 1.0, and 3.0% (*w*/*v*) bile salt (oxgall, MBcell, Korea) at 37 °C for 24 h. The surviving bacterial cells on MRS agar were enumerated and the survival rate was calculated as the percentage of colonies grown on MRS agar.

### 2.3. Antimicrobial Activity

Antimicrobial activity was examined using a paper disc assay according to Chiu et al. (2008). *Escherichia coli* KACC 13821, *Pseudomonas aeruginosa* KACC 10186, *P. putida* KACC 10266, *Bacillus cereus* KACC 12146, *Staphylococcus aureus* KCTC 1916, *S. epidermidis* KCTC1917, *S. xylosus* KACC 13239, and *Propionibacteria acnes* KCTC 3314 were used as indicator strains. Strains were grown in nutrient broth at 37 °C for 24 h, diluted to 10^8^ CFU/mL, and spread on a nutrient agar plate. Sterile discs were placed on the agar and 40 µL of neutralized (1 N NaOH) cell-free supernatants of lactobacilli strains were dropped. The diameter of each clear zone was determined after incubation at 37 °C for 12 h.

### 2.4. Cell Surface Hydrophobicity and Autoaggregation

Bacterial adhesion to hydrocarbons was determined according to the method described by Rosenberg (1984), with slight modifications as follows. After cultivation in MRS broth at 37 °C for 24 h, the obtained fresh cells were centrifuged (8000× *g*, 5 min) and washed three times with sterilized PBS (pH 7.2). The cells were resuspended and diluted in sterilized PBS to reach an optical density (OD) of 0.5, at 600 nm (A_0_), and 1.5 mL of the suspension was mixed with an equal volume of n-hexadecane (Sigma, St. Louis, MO, USA) in duplicate, followed by thorough vortex-mixing. The phases were allowed to separate for 1 h at room temperature, after which the aqueous phase was carefully removed and absorbance at 600 nm was measured (A_1_). The surface hydrophobicity (%) was calculated as follows: H% = (1 − A_1_/A_0_) × 100. The hydrophobicity was determined in triplicate. Cells with a hydrophobic index greater than 70% were arbitrarily classified as hydrophobic. For the autoaggregation assay, cells were grown for 18 h at 37 °C in MRS broth and washed three times with sterilized PBS (pH 7.2), resuspended, and diluted in sterilized PBS to reach an OD600 of 0.5 (A_0_). Four-milliliter aliquots of the cell suspensions were vortexed for 10 s, and autoaggregation was determined during 2 h of incubation at room temperature (At) in triplicate. The autoaggregation (%) was calculated as follows: A% = (1 − A_1_/A_0_) × 100.

### 2.5. Caenorhabditis elegans Intestinal Adhesion

Colonization of the *C. elegans* intestinal tract by PA-M76 was determined by measuring the number of bacterial cells in the worm intestines [16]. The *C. elegans* CF512 *fer-15(b26)II; fem-1(hc17)IV* strain was used in this study, which was routinely maintained on nematode growth medium (NGM) plates seeded with *E. coli* OP50. First, PA-M76 was sub-cultured (~1 × 10^9^ CFU/mL) three times before use. After exposing *C. elegans* to PA-M76 on NGM plates containing nystatin for 5 days, 10 worms were randomly picked, washed twice with M9 buffer, and placed on brain-heart infusion (BHI) plates containing kanamycin and streptomycin. Then, these plates were exposed to gentamycin (5 μL of a 25 μg/mL solution) for 5 min. Next, the worms were washed three times with M9 buffer and then pulverized using a pestle (Kontes Glass Inc., NJ, USA) in a 1.5 mL Eppendorf tube containing M9 buffer supplemented with 1% Triton X-100. After serial dilution in M9 buffer, the worm lysate was plated on MRS agar (pH 5.0), incubated for 48 h at 37 °C, and then counted for live bacterial cells. The results were compared with those of LGG used as a positive control.

### 2.6. In Vitro Immuno-Stimulatory Effects

#### 2.6.1. Cell Culture and Bacterial Stimulation

The murine macrophage cell line RAW 264.7 was purchased from the Korean Cell Line Bank (KCLB, Seoul, Korea) and was cultured in Dulbecco’s modified Eagle’s medium (DMEM) (HyClone, Marlborough, MA, USA) supplemented with 10% fetal bovine serum (FBS) and 1% penicillin–streptomycin solution (HyClone, Marlborough, MA, USA). The cells were incubated under 5% CO_2_ at 37 °C in a humidified incubator. For cytokine analysis, heat-treated LAB (110 °C, 30 min) were added at concentrations of 1 × 10^7^ CFU/well in RAW 264.7 cells after which the cells were plated at a density of 5 × 10^5^ cells onto a 96-well plate and cultured for 2 h. As a positive control for cell stimulation, 1 μg/mL lipopolysaccharide from *E. coli* (LPS; Sigma, St. Louis, MO, USA) was also used. After incubation for 24 h, RAW 264.7 cells with heat-killed LAB or LPS were harvested, and the cell culture medium was collected for real-time reverse transcription-polymerase chain reaction (qRT-PCR) assay.

#### 2.6.2. Cell Viability Assay

Cells were cultured in 96-well plates at 5 × 10^5^ cells/well for 2 h and subsequently treated with heat-killed LAB or LPS (1 µg/mL) for a further 24 h. Cell viability was then measured using the EZ-Cytox Assay Kit (Daeil Laboratory, Seoul, Korea) according to the manufacturer’s instructions. LPS was used as a positive control.

#### 2.6.3. qRT-PCR Analysis

Following each indicated treatment, total RNA was extracted from stimulated RAW 264.7 cells using the NucleoSpin RNA Kit (Macherey-Nagel, Duren, Germany), and cDNA was synthesized from RNA (2 μg) using the ReverTra Ace qPCR RT Master Mix (Toyobo, Osaka, Japan). PCR was performed using a CFX96 Touch Real-Time PCR Detection System (Bio-Rad, Hercules, CA, USA) with the SYBR Green Realtime PCR Master Mix (Toyobo, Osaka, Japan) in a 20 μL reaction volume. The target genes for the PCR were as follows: glyceraldehyde-3-phosphate dehydrogenase (*GAPDH*), tumor necrosis factor-alpha (*TNF-α*), interleukin-1 beta (*IL-1β*), *IL-6*, and *IL-12*. The primer sequences were as follows (5′-3′): *GAPDH* (forward CATGGCCTTCCGTGTTCCTAC; reverse TCAGTGGGCCCTCAGATGC), *TNF-α* (forward AGGCACTCCCCCAAAAGATG; reverse CACCCCGAAGTTCAGTAGACAGA), *IL-1β* (forward TGACGGACCCCAAAAGAT; reverse GTGATACTGCCTGCCTGAAG), *IL-6* (forward CCGGAGAGGAGACTTCACAGAG; reverse TCATTTCCACGATTTCCCAGAG), and *IL-12* (forward CGTGCTCATGGCTGGTGCAAAG; reverse CTTCATCTGCAAGTTCTTGGGC).

### 2.7. Fermentation of BRE with PA-M76

Organically grown BR were harvested from Gochang province in South Korea, quickly rinsed with tap water, and soaked in a jar with 40% (*w*/*v*) sugar at room temperature for 30 days at 20 °C. After soaking, the black raspberries were strongly pressed for 30 min. The obtained BRE was immediately diluted up to 20, 30, and 40° Brix with distilled water and then pasteurized for 15 min at 75 °C. The sterilized BRE was adjusted to pH 4–9 with 0.1 M sodium bicarbonate and 0.5 M citric acid (Samchun Pure Chemical Co. Ltd., Seoul, Korea). The strain PA-M76 was cultured in MRS broth for 24 h at 37 °C and then collected by centrifugation (1580MGR; Gyrozen, Daejeon, Korea) at 10,000× *g* for 15 min and washed twice with sterilized water. PA-M76 (*v*/*v*) was inoculated with BRE, and fermentation was conducted at 25–30 °C for 3–15 days. The fermented BRE (fBRE) samples were collected aseptically every 24 h for further analysis. The obtained fBRE was stored at −80 °C until further analysis.

### 2.8. Analysis of Cell Biomass and EPS

The obtained fBRE samples (20 mL) were boiled for 20 min to inactivate EPS-degrading enzymes and then quickly cooled to room temperature. Trichloroacetic acid was added to the fBRE samples at a final concentration of 4% (*w*/*v*), and then the samples were centrifuged at 8000× *g* for 20 min (1580 MGR, Gyrozen, Daejeon, Korea). Cell biomass was determined by measuring the weight of the precipitated pellet after drying in a hot air oven at 90 °C to a constant weight. Crude EPS were retrieved by ethanol precipitation after mixing three volumes of cold ethanol and stirring vigorously. The precipitated crude EPS formed overnight at 4 °C were collected by centrifugation and dried as described above. The remaining ethanol was dried in a vacuum concentrator. The dry weights of the cell biomass and EPS were expressed in g/L.

### 2.9. Total Anthocyanin, Phenol, and Flavonoid Analysis

The total anthocyanin content in fBRE was determined using the pH-differential method described by Giusti and Wrolstad (2001) in duplicate, with slight modifications. Briefly, samples were diluted sequentially up to 50-fold with 0.025 M potassium chloride buffer (pH = 1.0, Kanto Chemical Co., Inc., Tokyo, Japan) or 0.4 M sodium acetate (pH = 4.5, Samchun Pure Chemical Co. Ltd., Seoul, Korea) solution, and the absorbance of the mixture was measured at 530 and 700 nm using a UV-Vis spectrometer. The difference in absorbance between pH values and wavelengths was calculated (mg/L) using the following formula: (A × MW × DF × 1000)/(ε × 1), A: Absorbance of the diluted sample = (A_520nm_ − A_700nm_) pH 1.0 − (A_520nm_ − A_700nm_) pH 4.5, MW: molecular weight of cyanidin-3-glucoside = 449.2, ε: molecular absorptivity = 26,900, DF: dilution factor = 50. The total phenolic content in fBRE was determined according to the Folin–Ciocalteu method described by Ju et al. (2009). Gallic acid (Sigma Chemical Co., St. Louis, MI, USA) was used as a standard and each fBRE was diluted 50-fold with distilled water. Twenty-five microliters of diluted fBRE was transferred to a 1.75 mL tube to which 75 µL of 2 N Folin–Ciocalteu’s phenol reagent (Sigma Chemical Co., MI, USA) was added, and the mixture was maintained for 5 min at room temperature. This was followed by the addition of 200 µL of 7.5% sodium carbonate (Samchun Pure Chemical Co. Ltd., Seoul, Korea) solution and 700 µL of distilled water, and the mixture was incubated for 60 min at room temperature after which the absorbance was measured at 765 nm. The results were expressed as milligrams of gallic acid equivalent (GAE) per gram of fBRE. The total flavonoid content was determined using the modified Meda method (2005). Each fBRE was diluted 50-fold with distilled water. Five hundred microliters of diluted fBRE were transferred to a 1.75 mL tube to which 30 µL of 5% sodium nitrite (Junsei Chemical Co., Osaka, Japan) was added, and the mixture was incubated at room temperature for 5 min. The pre-reacted sample was added to 30 µL of 10% aluminum chloride (Fluka Chemical Co., NY, USA), and maintained at room temperature for 6 min. Finally, 200 µL of 1 M sodium hydroxide (Samchun Pure Chemical Co. Ltd., Seoul, Korea) was added to the maintained sample after which absorbance was measured at 510 nm. The total flavonoid content was determined using a catechin standard and the results were expressed as milligrams of catechin equivalent (CE) per gram of fBRE.

### 2.10. In Vitro Antioxidant Activity

#### 2.10.1. DPPH Radical Scavenging Activity

The antioxidant activity of fBRE was assayed by its scavenging effect on 2, 2-diphenyl-1-picrylhydrazyl (DPPH) as a free radical. Briefly, a 0.2 mM DPPH solution was prepared with methanol. Each sample was diluted with distilled water and then 100 µL of the diluted sample was mixed with 400 µL of prepared DPPH solution in a 1.75 mL tube. The mixture was covered with aluminum foil to protect it from light and maintained at 37 °C for 30 min. After the reaction, the absorbance of the resulting solution was measured at 517 nm against a blank. Ascorbic acid was used as a standard, and the percentage of DPPH quenched was calculated every minute as follows: % DPPH quenched = [1 − (Abs (sample)/(Abs (control) × 100].

#### 2.10.2. Lipid Peroxidation Inhibitory Activity

The lipid peroxidation inhibitory activity of BRE and fBRE was also measured according to the method described by Osawa and Namiki [17], with some modifications. Each sample was diluted with distilled water, and then 1 mL of the diluted sample was mixed with 1 mL of linoleic acid (50 mM), previously dissolved in ethanol (99.5%). After incubation at 60 °C in the dark for 8 days in tightly sealed glass vials, 100 μL of the oxidized BRE and fBRE samples were mixed with 4.7 mL of 75% (*v*/*v*) ethanol, 0.1 mL of 30% (*w*/*v*) ammonium thiocyanate, and 0.1 mL of 0.02 M ferrous chloride, dissolved in 1 M HCl. After 3 min, the absorbance of the oxidized samples was measured spectrophotometrically at 500 nm. Ascorbic acid was used as the standard and the inhibition degree of linoleic acid autoxidation percentage was calculated as follows: [(100 − sample Abs)/negative control Abs] × 100.

#### 2.10.3. Radical Cation ABTS+ Scavenging Activity

The radical cation (2,2′-azino-di-[3-ethylbenzthiazoline sulphonate]) (ABTS^+^) scavenging capacity of BRE and fBRE was measured using the Antioxidant Assay Kit CS0790 (Sigma Chemical Co., MI, USA), following the manufacturer’s instructions. Trolox (6-hydroxy 2,5,7,8-tetramethylchroman-2-carboxylic acid) was used as the antioxidant standard. The scavenging activity was expressed as the Trolox equivalent.

### 2.11. Headspace Solid-Phase Microextraction (SPME) and Gas Chromatography-Mass Spectrometry (GC-MS) Analysis

The aroma components were analyzed using SPME. To capture volatile compounds, 1 mL of culture, 1.5 g of NaCl, and 1 ppm of 2-methyl-3-heptanone (internal standard) were added to 20 mL glass vials (Supelco, Bellafonte, PA, USA). The SPME fiber needle was inserted into the vial, the volatile components were adsorbed on the autosampler for 30 min, and desorption was conducted using a GC-MS (7890B GC System/5977B MSD, Agilent Technologies) injection port at 220 °C for 5 min. Divinylbenzene/carboxen/polydimethy siloxanelsiloxane (DCP, 50/30 μm) was used as the SPME fiber. Each adsorbed volatile compound was analyzed using an HP-5 MS column (Agilent Technologies, Santa Clara, CA, USA; 30 m × 250 μm × 0.25 μm). The mobile phase gas was helium, and the flow rate was maintained at 1.0 mL/min. The oven temperature was maintained at 40 °C for 1 min and then increased to 250 °C at a rate of 20 °C/min and maintained for 3 min. The inlet temperature was 220 °C, and GC-MS was performed in the splitless mode with an MS ionization voltage of 70 eV, a source temperature of 230 °C, and an interface temperature of 280 °C. Mass spectra of each peak component separated by GC-MS were confirmed with those of the Mass Hunter database library (Agilent Mass Hunter Software, Agilent technology, USA). The volatile components were quantified by comparing the peak area of 2-methyl-3-heptanone and the peak area of the volatile flavor component identified.

### 2.12. Statistical Analysis

Data are expressed as the mean ± standard deviation (SD). Statistical significance was determined by one-way analysis of variance (ANOVA) using SPSS software version 21 (IBM SPSS Institute, Inc., Chicago, IL, USA), followed by Duncan’s test. Differences were considered statistically significant at *p* < 0.05.

## 3. Results and Discussion 

### 3.1. Fundamental Probiotic Properties: Tolerance and Antimicrobial Activity

To affect the GI, any potential probiotic strain should exhibit tolerance to acid and bile salts. After being subjected to gastric acidity (low pH) and intestine conditions (bile salts), the ability of probiotic strains to survive in adequate numbers is important for food industry applications [18]. In this study, the ability of PA-M76 isolated from Korean fermented rice wine to survive in simulated gastric juice and bile salts was evaluated in vitro. As shown in Table 1, the growth of PA-M76 decreased (*p* < 0.05) after 3 h of incubation at 37 °C under acidic conditions (pH 2.5), but showed a survival rate of 71.7%. The level was similar to the survival levels of PS and LGG, in which the control and reference probiotic strains exhibited survival percentages >70% 3 h after exposure to simulated intestinal juice (*p* > 0.05). Considering that the low pH of 1.5 to 3.0 in the human stomach may increase to 4.5 during ingestion, PA-M76 may achieve a higher survival ability under low pH in the stomach. Compared to our strain, other *Pediococcus* strains could retain their ability at pH 3.0–4.0, but strains displayed high viability loss at lower pH. Furthermore, PA-M76 strains were highly resistant to bile stress. No differences in the cell counts were observed when PA-M76 strains were incubated for 24 h with bile salts ranging from 0.3 to 3.0% (Table 1). In contrast, compared with PA-M76, the reference strains PP and LGG showed marginal viability loss, with survival rates of 89.1 and 80.9% under 3.0% bile salt conditions, respectively (*p* < 0.05). The human intestine contains the relevant physiological concentrations of bile salts, ranging from 0.3 to 0.5%, and bile salts are harmful to living cells because they damage the structure of the cell membrane [8]. In the strain selection assay [19], those with growth rates above 50% showed good resistance in the presence of 0.3% bile salts. In this regard, the high survival ability of PA-M76 might reflect good resistance to bile and the low pH of gastric juice, which indicated that the strain could survive under harsh conditions in the gut. Additionally, compared to control strains, PA-M76 showed a broader inhibitory activity spectrum against most of the tested pathogens, regardless of the presence of Gram-negative and Gram-positive bacteria, as shown in Table 2.

Regarding results of the tests for pH, temperature, NaCl and glucose tolerance, numbers indicates that the test strain can grow on the MRS broth with the respective conditions. Survival rates (%) of *P. acidilactici* M76 in simulated gastric juices and bile salt. Different small letters (a–c) in the same column indicate significant differences of values among three different LAB strains (*p* < 0.05). PA-M76, *P. acidilactici* M76; PP-33316, *P. pentosaceus* (ATCC 33316); LGG, *Lactobacillus rhamnosus* GG (ATCC 53103). Viable cell numbers were counted on MRS agar after incubation at 37 °C for 48 h.

### 3.2. Adhesion Properties of PA-M76

The ability to adhere to the intestinal epithelial surface and colonize the GIT is one of the main desirable characteristics of probiotics [20]. To determine the adhesive properties of PA-M76, we first examined its surface hydrophobicity and autoaggregation. As shown in Figure 1A, PA-M76 showed unexpectedly lower surface hydrophobicity and autoaggregation than LGG (73.5 and 76.3% versus 61%, respectively; *p* < 0.05), whereas PP showed lower hydrophobicity and autoaggregation compared to the other two strains. Recently, as an alternative to in vitro models, such as propagated human intestinal cell lines, a *C. elegans* surrogate in vivo model was successfully used as a simple, rapid, and economic model system to study bacteria–host interactions in the gut, as the intestinal cells of *C. elegans* are similar in structure to those in humans [21]. In this study, a *C. elegans* model system was used to investigate the ability of PA-M76 to attach to the intestinal tract. As shown in Figure 1B, PA-M76 showed high GIT colonization ability. The strain exhibited outstanding persistence in the *C. elegans* intestine with a cell count of over 5.0 log CFU/mL per worm for 5 days. Unexpectedly, LGG, used as the control strain in this study, showed poor attachment in the gut environment at 1 CFU/mL per worm, although LGG has been shown to bind to enterocytes in a previous study [22]. Four strains of *Lactobacillus* sp. were selected as potential probiotic bacteria among 2000 LAB strains isolated from infant feces because of their remarkably high colonization ability on the *C. elegans* intestine of over 4.3 CFU/mL per worm after 5 days [20]. Furthermore, *Pediococcus* strains exhibited relatively high GIT colonization (>3.5 log CFU/mL per worm), whereas other *Lactobacillus*, *Streptococcus*, and *Weissella* sp. could not colonize the intestinal tract of *C. elegans* [23]. Moreover, EPS synthesized by several bacteria play a role in adhesion to surfaces, such as eukaryotic cells, and the modulation of the host immune system [2,20]. Thus, the results indicate that PA-M76 shows excellent colonization of the worm intestinal tract, probably due to its ability to produce EPS [24].

### 3.3. Immuno-Stimulatory Activity of PA-M76 to RAW 264.7 Cells

Intestinal macrophages are key players in mucosal immune responses to defense against injurious agents, such as pathogens and food antigens. Some LAB strains can activate innate immune cells by inducing or enhancing cytokine production [25]. Probiotics may exert immunomodulatory effects in a strain-specific manner via specific bacterial components [26]. Therefore, each probiotic candidate must be evaluated to identify its specific biological activity. In this study, the immune-stimulatory activity of PA-M76 on RAW 264.7 macrophages was assessed. First, the cytotoxic effects of the M76 strain and reference strains against RAW 264.7 cells were determined using the MTT assay, and none of the strains had any observed effect on the viability of RAW 264.7 cells after treatment for 24 h (data not shown). To assess the effects of PA-M76 on immune responses, the expressions of cytokine genes, such as *IL-1β*, *IL-6*, *IL-12*, and *TNF-α*, which may be produced by the activated macrophages, were determined qRT-PCR. Our results showed that the production of four cytokines in RAW 264.7 cells was significantly increased by treatment with LAB strains (Figure 2). Furthermore, treatment with PA-M76 resulted in significantly higher expression of *IL-1β*, *IL-6*, and *IL-12* than the control strains PP and LGG (*p* < 0.05). Additionally, the expression of TNF-α by PA-M76 was almost similar to that by LGG. To date, various probiotic LAB strains have been shown to enhance nonspecific cellular immune responses, including activation of macrophages, natural killer (NK) cells, antigen-specific cytotoxic T-lymphocytes, and the release of various cytokines [26]. These health effects imparted by probiotic bacteria are strain-specific but not intra-strain specific. Many studies on probiotic cultures have been conducted for *Lactobacillus* and *Bifidobacterium* sp., but only a few have reported the immunomodulatory activities of *Pediococcus* sp., such as *P. pentosaceus* NB-17, originated from Japanese traditional vegetable pickles [27], *P. parvulus* 2.6, isolated from cider [19], *P. pentosaceus* OZF, derived from human breast milk [25], and *P. pentosaceus* L1, isolated from Chinese fermented vegetables [28]. Moreover, little is known about the immunomodulatory activities of *P. acidilactici*. Our study showed that the treatment of PA-M76 induced the expression of cytokine genes in RAW 264.7 cells, and, to our knowledge, this is the first study to report the effects of *P. acidilactici* on immunomodulatory function.

### 3.4. Black Raspberry Fermentation by PA-M76 Enhanced Production of Functional EPS

The BRE did not allow the growth of PA-M76 at a low initial inoculation cell density. The initial inoculation cell density of PA-M76 increased up to 8.2 log CFU/mL, which was maintained during fermentation (data not shown here). As shown in Figure 3A, when the BRE was fermented with PA-M76 at 25 °C, the cell density of the starter increased until 48 h and decreased slightly up to 15 days of fermentation. However, PA-M76 cell density increased very slowly up to 72 h above 30 °C and decreased dramatically at the late fermentation stage, as shown in Figure 3A. The value of pH decreased to 3.53 ± 0.12 and 3.47 ± 0.09, respectively, at these temperatures after 15 days of fermentation. Generally, mesophilic bacteria of *Pediococcus* species, including *Lactococcus*, *Leuconostoc* species, *Lactobacillus kefir*, *L. brevis L. fermentum*, and *Bifidobacterium bifidum*, showed an almost 50% higher EPS production rate when the organisms were grown at a lower temperature of 25 °C than at 30 °C [29]. The cell biomass of PA-M76 markedly increased at pH 6 and 40°Brix in a sugar concentration-dependent manner. However, the concentration of EPS did not correlate significantly with the increase in cell biomass (Appendix A). In contrast, EPS production by strain M76 in BRE was more favorable at a higher acidic pH (4.0) than at a neutral pH (6.0–7.0), although the production of EPS tended to increase in a sugar concentration-dependent manner. The physiological factors that play a crucial role in EPS production include pH, temperature, incubation time, and medium composition. Particularly, acidic stress usually inhibits bacterial growth but can stimulate EPS production in some LAB strains [30]. The high acidity of the BRE might cause acid stress to the cells, resulting in increased production of EPS [31]. In this study, EPS production by PA-M76 showed a time-dependent dramatic increase in synthesis rate until 72 h, producing 1.62 g/L as shown in Figure 3B (*p* < 0.05). EPS production tended to decline after 72 h, suggesting enzymatic degradation caused by prolonged cultivation, but it was still maintained up to 1.04–1.08 g/L until the final stage of BRE fermentation. The cell biomass was highest at the initial stages on days 3 and 15 (3.14 and 3.35 g/L, respectively), but EPS production was highest on day 3 (1.62 g/L). As a result, maximum EPS production by PA-M76 was observed after three days of BRE fermentation. It seemed that extending the fermentation time resulted in decreased EPS content due to increased EPS degradation activity. To our knowledge, the volume of EPS produced under these optimized conditions in BRE is comparatively sufficient to prepare functional beverages because of its observed proliferative effect on RIN-m5F cells, cytoprotective activity against alloxan-induced cytotoxicity at 10 mg/mL concentration, and lipid-lowering effect at 10 mg/mL concentration. In conclusion, maximum EPS production was achieved up to 1.62 g/L under optimal conditions at pH 4.0, at 25 °C for 3 days with 30°Brix of BRE. This volume of EPS produced under the optimized conditions in fBRE would be sufficient to produce functionally active beverages with health-promoting effects.

### 3.5. Phenolic Compounds and Antioxidant Activity of fBRE

We evaluated the contents of phenolic compounds such as total anthocyanins (TA), total polyphenols (TP), and total flavonoids (TF), of fBRE fermented under the optimized conditions with high EPS production, as described above. As shown in Table 3, the TA concentration of fBRE (45 TAC mg/L) was almost identical to that of NfBRE (43 TAC mg/L), suggesting that the lactic acid fermentation of BRE by PA-M76 was not significantly different from that of non-fermented BRE (NfBRE). Analysis of TF also showed no significant (*p* < 0.05) difference between fBRE and (99.9 mg CE/100 g) versus 81 mg CE/100 g in NfBRE and fBRE, respectively). However, a significant (*p* < 0.05) increase was only found at TP (379.3 mg GAE/100 g) compared to NfBRE (349.4 mg GAE/100 g). Lactic acid fermentation of berries is challenging owing to their high contents of acids and phenolic compounds. The abundant phenolic compounds in berries, including hydroxycinnamic, neochlorogenic, and chlorogenic acids, are usually hydrolyzed to quinic and caffeic acids, which are subsequently degraded to vinyl and ethyl catechols via the action of phenolic acid decarboxylases and reductase [32]. However, reportedly, mulberry juice fermented with *L. plantarum*, *L. paracasei*, and *L. acidophilus* showed an increase in the content of phenolic acids, total anthocyanins, and flavanols [33]. The effect of lactic acid fermentation might be debatable, but the phenolic compound profile during lactic acid fermentation is greatly affected by the strain and starting materials used. β-Glucosidase enzymes can hydrolyze the flavonoid conjugates during lactic acid fermentation and influence polyphenols during lactic acid fermentation [34]. The strain used in this study, PA-M76, might not be sufficient for enhancing the content of phenolic compounds during lactic acid fermentation of BRE, even though TP increased slightly, possibly due to this catalyst.

Three different methods were used to assay the antioxidant activity in vitro with NfBRE and fBRE after fermentation with *P. acidilactici* M76. First, the antioxidant activity was assayed by DPPH radical scavenging activity using the oxidant sTable 2,2-diphenyl-1-picrylhydrazyl (DPPH) radical. The DPPH radical scavenging activity of fBRE was significantly higher than that of NfBRE and showed a radical scavenging activity of 51.9% (*p* < 0.05) towards the stable radical DPPH, as shown in Figure 4A. We also measured the antioxidant activity by assaying the quantification of the inhibition of linoleic acid peroxidation, which is thought to proceed through radical-mediated abstraction of hydrogen atoms from methylene carbons in polyunsaturated fatty acids [35]. When fBRE was assayed, the rate of inhibition of linoleic acid (44.8%, *p* < 0.05) was significantly higher than that of NfBRE, as shown in Figure 4B. We also used a radical cation, ABTS (2,2′-azino-di-[3-ethylbenzthiazoline sulfonate] to determine the total antioxidant activity of fBRE and NfBRE. When fBRE was used for this assay, the ABTS scavenging activity of the fBRE was 23.2 ± 1.3 mM, which was significantly higher than that of NfBRE as shown in Figure 4C. Interestingly, the antioxidant activity of radical scavenging activity, inhibition activity of linoleic acid peroxidation, and total antioxidant capacity towards ABTS radical of the fBRE by PA-M76 were significantly higher than those of NfBRE, indicating enhanced lactic acid fermentation by PA-M76. Previously, the crude EPS extract from PA-M76 showed the highest DPPH radical scavenging activity of 48.1% at a concentration of 1 mg/mL, with a dose-dependent increase in the activity [24]. The fermentation of BRE by PA-M76 caused an increase in the concentration of EPS (1.62 mg/L), as shown in Figure 3B. Although black raspberries are generally known as an abundant source of phytochemicals, such as phenolic acids, flavonoids, anthocyanins, and tannins, with well-documented antioxidant activity, we did not observe a significant increase in the contents of those phenolic compounds in this study, as described above. No significant differences in the total antioxidant activity were observed between NfBRE and fBRE. Therefore, the antioxidant potential of NfBRE and fBRE can be largely attributed to the originally existing phenols in the raw BRE. Nevertheless, our results showed that the significantly enhanced antioxidant properties of fBRE might be caused by EPS-enriched fermentation by PA-M76.

### 3.6. Production of Volatile Compounds during Fermentation of BRE with P. acidilactici M76

During fermentation with PA-M76, a total of 28 volatile compounds were identified, as shown in Table 4. Terpenes (7) and esters (12) were present in higher numbers than acids (3), alcohols (5), and aldehydes (1), which are the commonly identified volatile compounds in fruit-based fermented products. The total concentration of volatile compounds in fBRE was higher than that in NfBRE. During fermentation, esters increased in concentration and expanded as the largest group of compounds in which ethyl acetate (fruity/sweet-like odor), ethyl octanoate (fruity/winey/sweet-like order), and ethyl decanoate (sweet waxy/fruity) were detected as the major compounds [36,37]. Importantly, these compounds are considered the highest odor-active compounds in fermented fruit beverages [38,39]. In the case of fermented beverages, isoamyl acetate and ethyl hexanoate, which have small molecular weights, mainly contribute to the fruit flavor. Esters represent a major volatile component in fermented fruit beverages [39]. In this study, ethyl acetate, which produces fruity and sweet-like notes, accounted for 29% of the total volatile compound abundance at the final stage, which was not detected at the initial stage of fermentation and in NfBRE. In the alcohol groups, isoamyl alcohols (fruity/sweet-like odor) dramatically increased in concentration as the largest group of compounds during BRE fermentation, which is usually high in wine-producing flowers and honey smells [40]. Additionally, phenylethyl alcohol (floral/rose) was detected as the major alcohol, and its amount increased with increasing fermentation time from 0.05 ± 0.02 to 0.64 ± 0.31. This volatile compound was one of the major potentially significant higher alcohols from grapes, which may be synthesized in wine as a by-product of yeast fermentation. Their synthesis closely parallels that of ethanol production. Furthermore, a similar number of compounds, including phenylethyl alcohol, accounted for approximately 50% of the aromatic constituents of wine. However, benzaldehyde (burnt bitter/sharp), which exists in NfBRE, completely disappeared by the end of the fermentation period. Cherry propanol (sweet; fruity/cherry) was detected as one of the volatile compounds in NfBRE that gradually decreased and disappeared after fermentation. Terpenes, such as terpinen-4-ol (woody flavor) and myrtenol (medicinal and mint flavor), which were relatively higher in NfBRE [41,42] and considered as undesirable flavors for fermented products, disappeared or decreased during BRE fermentation. Acid components, such as acetic, octanoic, and decanoic acids, generally impart a poor rancid flavor to foods [43]. Acids such as acetic and octanoic acids remained until the final stage of fBRE, but the amount was too small to be considered.

To understand the interrelationships between volatile flavor compositions and fBRE depending on the fermentation stage, PCA analyses were performed. As shown in Figure 5, an overall PCA biplot was constructed with a total variance of 65.8%. PC1 showed a strong positive correlation to fBRE components and strong negative correlation to NfBRE components. Additionally, PC2 loading clearly showed that the fBRE at different stages was clearly differentiated, with fBRE-2 and fBRE-3 positioned fully in the strongly positive PC1 region due to increased concentrations of esters and alcohols. The later-stage samples were also positioned in the positive PC1 region because of the elevated concentrations of esters. It can be noted that the fBRE tended to have a high concentration of most volatile compounds.

## 4. Conclusions

In this study, functional EPS-producing PA-M76 was shown to resist biological barriers, such as simulated gastric and pancreatic juices, particularly colonizing the surface of the *C. elegans* intestinal tract in higher populations and exhibiting significant in vitro immunostimulatory activity compared to the conventional commercial starter LGG. The results of the current investigation revealed promising perspectives for the application of functional EPS-producing PA-M76 as a functional probiotic. Moreover, we also applied the EPS-producing strain PA-M76 to prepare functionally enhanced black raspberry beverages and showed how the antioxidant properties of black raspberry extract could be enhanced through lactic acid fermentation by this strain. Considering the great interest in natural materials with functional ingredients, the health-promoting effects of PA-M76 can be exploited for the bioprocessing of diverse food materials in the food industry.

## Figures and Tables

**Figure 1 microorganisms-09-01364-f001:**
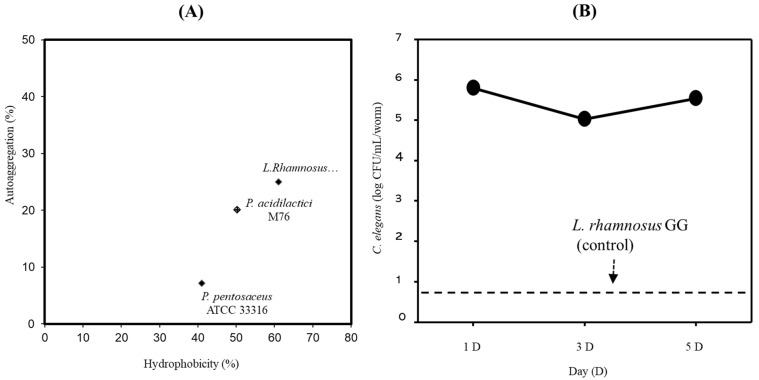
(**A**) Correlation of the hydrophobicity and coaggregation results of the lactic acid bacteria strains. (**B**) Adhesion activity of *Pediococcus*
*acidilactici* M76 in the intestine of *Caenorhabditis elegans*.

**Figure 2 microorganisms-09-01364-f002:**
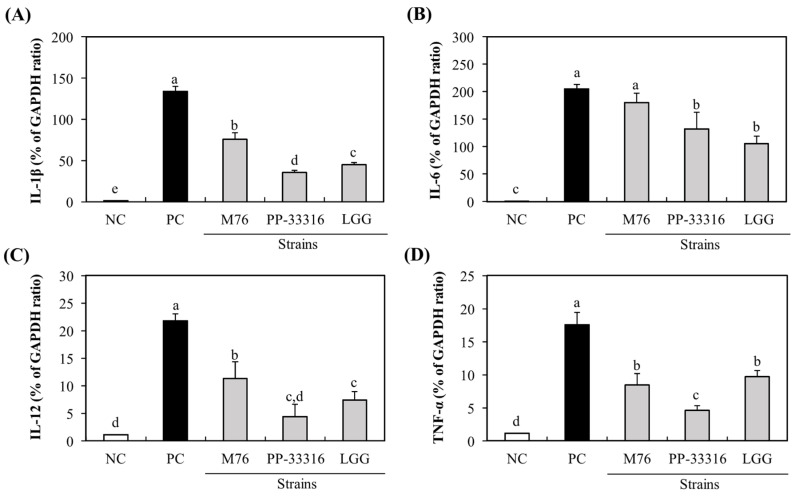
Effect of *Pediococcus acidilactici* M76 on mRNA expression levels of macrophage-activating factors, such as *IL-1β* (**A**), *IL-6* (**B**), *IL-12* (**C**), and *TNF-α* (**D**). RAW 264.7 cells were treated with different lactic acid bacteria (LAB) strain fractions for 24 h. Cells treated with media alone were used as the negative control (NC), cells treated with lipopolysaccharide (LPS, 1 μg/mL) were used as the positive control (PC). PA-M76, *Pediococcus acidilactici* M76; PP-33316, *Pediococcus pentosaceus* (ATCC 33316); LGG, *Lactobacillus rhamnosus* GG (ATCC 53103). ^a–e^ Means with different letters are significantly different (*p* < 0.05).

**Figure 3 microorganisms-09-01364-f003:**
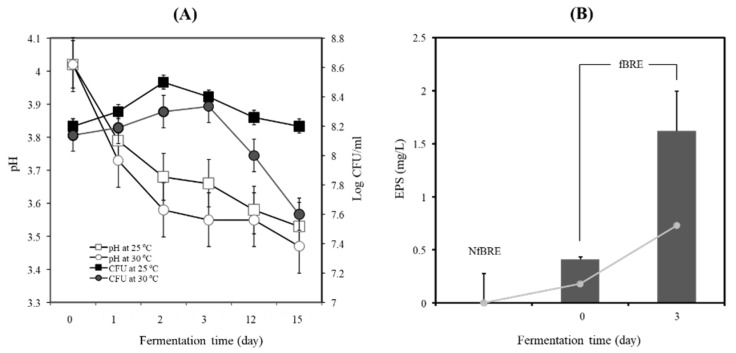
(**A**) Changes in pH, viable cell count, and (**B**) exopolysaccharide production of the fermented black raspberry extract by *Pediococcus acidilactici* M76. Cultivation was done for 15 days. log CFU/mL at 25 °C: (■), log CFU/mL at 30 °C: (●), pH at 25 °C: (□), pH at 30 °C: (○).

**Figure 4 microorganisms-09-01364-f004:**
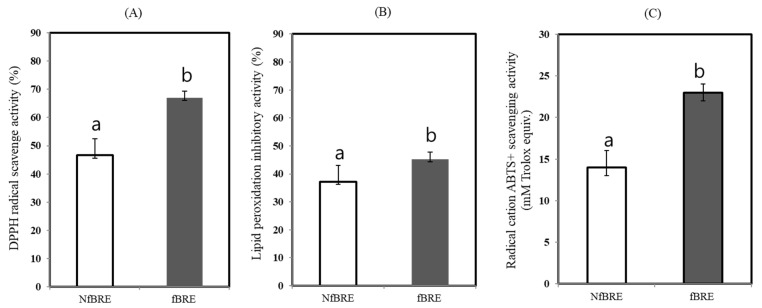
Changes in antioxidant activity of the fermented black raspberry extract by *Pediococcus acidilactici* M76. (**A**) DPPH radical scavenge activity (%). (**B**) Lipid peroxidation inhibitory activity (%). (**C**) Radical cation ABTS+ scavenging activity. NfBRE: (□), fBRE: (■). ^a,b^ Means with different letters are significantly different (*p* < 0.05).

**Figure 5 microorganisms-09-01364-f005:**
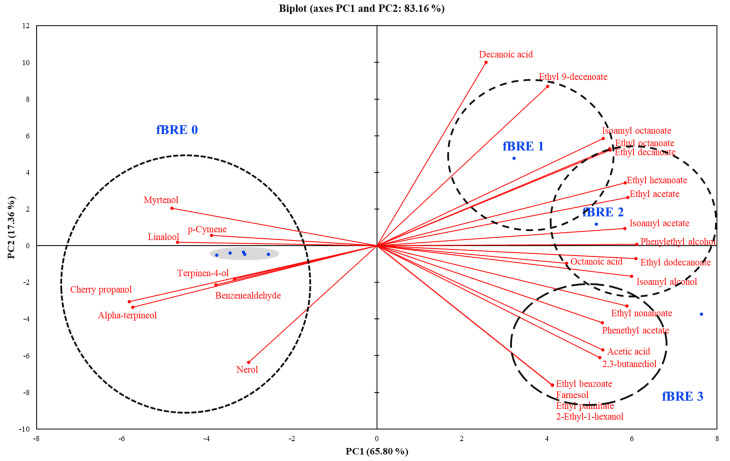
Biplot of the principal component analysis of volatile compounds from fermented black raspberry extract beverage inoculated with *Pediococcus acidilactici* M76.

**Table 1 microorganisms-09-01364-t001:** Physiological and biochemical properties of *Pediococcus acidilactici* M76.

Tolerance	PA-M76	LGG	*P. pentosaceus* 33316
pH 7.0	111.7 ± 0.58	110.7 ± 0.76	111.4 ± 0.35
pH 2.5	71.7 ± 0.22	72.5 ± 0.42	74.0 ± 0.08
Bile salt 0.3%	101.8 ± 0.92	97.9 ± 0.78	95.9 ± 0.61
Bile salt 1%	104.0 ± 1.11	93.3 ± 0.53	94.1 ± 0.78
Bile salt 3%	101.4 ± 0.15	80.9 ± 1.73	89.1 ± 1.33
Temperature	35~50	35~50	35~50
NaCl (%, *w/v*)	4~8	4~7	4~8
Glucose (%, *w/v*)	4~14	4~14	4~14

**Table 2 microorganisms-09-01364-t002:** Inhibitory activity of *Pediococcus acidilactici* M76 against pathogenic microorganisms.

Strain	Pathogenic Microorganisms									
	*S. aureus*	*S. epidermidis*	*S. xylosus*	*P. aeruginosa*	*P. putida*	*B. cereus*	*B. subtilis*	*B. vallismortis*	*E. coli*	*P. acnes*
*L. Rhamnosus* GG ***	++++	+	+++	++	++	++	+	+++	+	++
*P. acidilactici* M76	++++	+++	+++	+++	+++	++++	++	+++	++++	+++
*P. pentosaceus* ATCC 33316	++++	+	+++	++	+++	++	++	+++	+++	++

* *L.*
*rhamnosus* GG is used as control strain. ++++, inhibition zone > 4 mm; +++, inhibition zone > 3 mm; ++, inhibition zone > 2 mm; +, inhibition zone > 1 mm; <+, inhibition zone < 1 mm.

**Table 3 microorganisms-09-01364-t003:** Phenolic compounds of fermented black raspberry extract (fBRE).

Phenolic compounds	NfBRE	fBRE
Total polyphenols (mg GAE/100g)	349.4 ± 3.99 ^a^	379.3 ± 4.38 ^b^
Total flavonoids (mg CE/100g)	99.9 ± 2.37 ^a^	81 ± 0.82 ^a^
Total anthocyanins (TAC mg/L)	43 ± 0.67 ^a^	45 ± 0.14 ^a^

NfBRE: Non fermented black raspberry extract, fBRE: Fermented black raspberry extract; ^a,b^ Means with different letters are significantly different (*p* < 0.05).

**Table 4 microorganisms-09-01364-t004:** Change of volatile flavor compound contents during black raspberry fermentation by *Pediococcus acidilactici* M76.

Aroma Compound	RT ^(1)^	Non Fermented Black Raspberry Extract (Day)				Fermented Black Raspberry Extract (Day)				Odor Description ^(2)^
		0	1	2	3	0	1	2	3	
*Acids*										
Acetic acid	2.77	nd ^(3)^	nd	nd	nd	nd	nd	0.03 ± 0.01 ^ab^	0.05 ± 0.03 ^a^	Sour
Octanoic acid	6.63	nd	nd	nd	nd	nd	nd	0.18 ± 0.16 ^a^	0.08 ± 0.07 ^ab^	Fatty, waxy, cheesy
Decanoic acid	8.01	nd	0.01 ± 0.01	0.01 ± 0.00	nd	nd	0.32 ± 0.28	0.07 ± 0.06	0.02 ± 0.01	Rancid, sour, fatty, citrus
*Alcohols*										
Isoamyl alcohol	2.93	nd	nd	nd	nd	nd	0.65 ± 0.19 ^b^	1.48 ± 0.25 ^a^	1.75 ± 0.34 ^a^	Alcoholic, whiskey, fruity
2,3-Butanediol	3.62	nd	nd	nd	nd	nd	nd	0.01 ± 0.01 ^ab^	0.02 ± 0.01 ^a^	Fruity, creamy, buttery
2-Ethyl-1-hexanol	5.48	nd	nd	nd	nd	nd	nd	nd	0.05 ± 0.00	Citrus, floral, oily, sweet
Phenylethyl alcohol	6.25	0.02 ± 0.00 ^b^	0.02 ± 0.01 ^Ba^	0.02 ± 0.01 ^Ba^	0.02 ± 0.01 ^Ba^	0.05 ± 0.02 ^Ab^	0.40 ± 0.25 ^Aab^	0.48 ± 0.21 ^Aa^	0.64 ± 0.31 ^Aa^	Floral, rose, dried rose
Cherry propanol	6.79	0.15 ± 0.01	0.17 ± 0.03	0.16 ± 0.04	0.15 ± 0.02	0.12 ± 0.02	nd	nd	nd	Sweet, fruity, cherry
*Aldehyde*										
Benzealdehyde	4.95	0.01 ± 0.01 ^Ab^	0.03 ± 0.00 ^Aa^	0.04 ± 0.00 ^a^	0.04 ± 0.00 ^a^	nd	nd	nd	nd	Sharp, sweet, bitter, almond
*Esters*										
Ethyl acetate	2.11	nd	nd	nd	nd	nd	1.99 ± 0.30	1.60 ± 0.85	2.08 ± 0.54	Ethereal, fruity, sweet, grape
Isoamyl acetate	4.14	nd	nd	nd	nd	nd	0.30 ± 0.07 ^b^	0.63 ± 0.11 ^a^	0.49 ± 0.11 ^a^	Sweet, banana, fruity
Ethyl hexanoate	5.22	nd	nd	nd	nd	nd	0.15 ± 0.12	0.13 ± 0.10	0.14 ± 0.12	Sweet, fruity, pineapple
Ethyl benzoate	6.67	nd	nd	nd	nd	nd	nd	nd	0.05 ± 0.01	Fruity, sweet, wintergreen, minty
Ethyl octanoate	6.82	nd	nd	nd	nd	nd	1.11 ± 0.94	0.89 ± 0.73	0.75 ± 0.60	Fruity, winey, waxy, sweet
Phenethyl acetate	7.31	nd	nd	nd	nd	nd	nd	0.17 ± 0.15	0.18 ± 0.16	Floral, rose, sweet, honey
Ethyl nonanoate	7.53	nd	nd	nd	nd	nd	0.02 ± 0.02	0.04 ± 0.04	0.07 ± 0.05	Fruity, rose, waxy, rummy
Ethyl 9-decenoate	8.16	nd	nd	nd	nd	nd	0.50 ± 0.44	0.23 ± 0.20	0.13 ± 0.11	Fruity, fatty
Ethyl decanoate	8.21	nd	0.04 ± 0.05	0.03 ± 0.04	0.01 ± 0.02	nd	0.52 ± 0.44	0.45 ± 0.38	0.36 ± 0.30	Sweet, waxy, fruity, apple
Isoamyl octanoate	8.55	nd	nd	nd	nd	nd	0.05 ± 0.04	0.04 ± 0.04	0.03 ± 0.02	Sweet, oily, fruity, green
Ethyl dodecanoate	9.46	nd	nd	nd	nd	nd	0.05 ± 0.05	0.08 ± 0.06	0.10 ± 0.08	Sweet, waxy, floral, soapy
Ethyl palmitate	11.6	nd	nd	nd	nd	nd	nd	nd	0.01 ± 0.00	Waxy, fruity, creamy, milky
*Terpenes*										
p-Cymene	5.5	0.07 ± 0.01 ^a^	0.03 ± 0.00 ^Ab^	0.02 ± 0.02 ^b^	nd	0.05 ± 0.01 ^a^	0.02 ± 0.00 ^Bb^	nd	nd	Fresh, citrus, terpenic
Linalool	6.08	0.09 ± 0.01 ^a^	0.06 ± 0.01 ^Ab^	0.05 ± 0.01 ^Abc^	0.03 ± 0.00 ^Ac^	0.07 ± 0.02 ^a^	0.04 ± 0.01 ^Bb^	0.02 ± 0.01 ^Bb^	0.02 ± 0.00 ^Bb^	Citrus, floral, sweet
Terpinen-4-ol	6.76	nd	0.18 ± 0.01 ^a^	0.16 ± 0.02 ^b^	0.12 ± 0.01 ^c^	nd	nd	nd	nd	Peppery, woody, earthy
alpha-terpineol	6.85	0.35 ± 0.02 ^a^	0.34 ± 0.02 ^Aab^	0.31 ± 0.03 ^Abc^	0.27 ± 0.02 ^Ac^	0.29 ± 0.03 ^a^	0.14 ± 0.13 ^Bb^	0.13 ± 0.04 ^Bb^	0.15 ± 0.02 ^Bb^	Pine, terpenic, lilac, citrus
Myrtenol	6.91	0.18 ± 0.01 ^a^	0.16 ± 0.01 ^ab^	0.13 ± 0.01 ^b^	0.11 ± 0.01 ^c^	0.18 ± 0.02 ^a^	0.13 ± 0.06 ^ab^	0.10 ± 0.03 ^b^	0.08 ± 0.02 ^b^	Woody, pine, balsamic
Nerol	7.25	0.02 ± 0.00 ^a^	0.01 ± 0.00 ^b^	0.01 ± 0.00 ^c^	0.01 ± 0.00 ^a^	0.02 ± 0.00 ^a^	nd	0.01 ± 0.00 ^b^	0.01 ± 0.00 ^b^	Fresh, citrus, floral
Farnesol	10.24	nd	nd	nd	nd	nd	nd	nd	0.01 ± 0.01	Sweet, floral

Relative contents of volatile compounds were determined as peak areas of an internal standard (1 ppm of 2-methyl-3-heptanone in methanol). ^(1)^ RT, retention time ^(2)^ Odor descriptions were cited from http://www.thegoodscentscompany.com/ (accessed on 11 October 2020) ^(3)^ nd, not detected. Different capital (A and B) and small letters (a–c) in the same column indicate significant differences of values between two samples with same fermentation day and between eight samples with fermentation time and sample types, respectively (*p* < 0.05).

## Data Availability

Not applicable.

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
