# Peer review of "Evaluation of Probiotic Properties of *Pediococcus acidilactici* M76 Producing Functional Exopolysaccharides and Its Lactic Acid Fermentation of Black Raspberry Extract"

_microorganisms, 2021, doi:10.3390/microorganisms9071364_

Round 1

Reviewer 1 Report

microorganisms-1248035

The manuscript deals with the evaluation of probiotic properties of Pediococcus acidilactici M76 producing functional exopolysaccharides and its lactic acid fermentation on black raspberry extract. The manuscript presents valuable information and AA-M76 could be an interesting strain for commercial exploitation. However, the manuscript present abundant and misleading errors which should be corrected before publication. Therefore, a complete and careful revision by the authors is recommended. A no-exhaustive guide of points to clarify follows.

Specific notes:

L 18. The sentence requires explanation since it is unrealistic that PA-M76 could produce polyphenols.

L50. Italic scientific name.

L101. How was controlled the effect of just the lactic acid present in the supernatant?

L121 Please complete name in headings

L203-2012. Could the changes observed in total polyphenol be due to the reactions with different compounds in fermented and non-fermented extracts?

L244-245. Revise the sentence.

L274-275. Was considered the increase in error because of the multiple comparisons?

Table 1. Footnotes mention the presence of different small letters in the Table. Apparently, they are missing.

L327-329. Figure 1. No subgraphs are observed in Fig. 1.

L335. Again, there is confusion regarding the Figure they are referring to.

L373. There is another confusion regarding Figure.

L388. Apparently, there is confusion on the type of fermentation, although volatile are also produced.

L392. Please, refer to the appropriate Fig.

Figure 3 should be modified since the legend says “Fermentation time (day) but the intervals are named as fBRE. Please remove fBRe and explain that the graph is referred to fBRE in the legend of the figure.

L403-413. Explanation of EPS production should be clarified.

L449-450. The sentence should be clarified. Is PA-M76 able to synthesize phenols?

L488. There is confusion regarding the number of the table.

L486-518. Were considered for the comparisons the possible different volumes of the volatile present in the different samples analyzed.

L512 expressions such as “gradually decrease”, “increased” are not fully supported by the experimental results. Please accommodate the expressions to actual data.

L521. The total variance was wrong.

L522. The use of NBRE and fBRE0 can introduce confusion in the manuscript understanding.

Author Response

To reviewer

Thanks for your valuable comments and detail corrections on our manuscript. We revised our manuscript according to your comment as below.

Point1: L 18. The sentence requires explanation since it is unrealistic that PA-M76 could produce polyphenols.

Response 1: PA-M76 can not produce polyphenols. The fermentation of black raspberry extract by this strain might afftect polyphenol contents. It usually happen when lactic acid fermentation carried out on vegetable or fruits.

Point 2: Italic scientific name.

Response 2: Thanks. We corrected it.

Point 3: How was controlled the effect of just the lactic acid present in the supernatant?

Response 3: Antimicrobial activity usually carried out as described in our manuscript by using cell-free supernatant of lactobacilli strains.

Point 4: L121 Please complete name in headings

Response 4: We wrote it as full name of Caenorhabditis elegans. Thanks.

Point 5: L203-2012. Could the changes observed in total polyphenol be due to the reactions with different compounds in fermented and non-fermented extracts?

Response 5: Yes. In many studies, it was proven that polyphenols can be affected by lactic acid fermentation.

Point 6: L244-245. Revise the sentence.

Response 6: We revised the sentence. Thanks.

Point 7: L274-275. Was considered the increase in error because of the multiple comparisons?

Response 7: I am not sure I can understand about your question. But, we did statical analysis to compare the obtained data in triplicate.

Point 8: Table 1. Footnotes mention the presence of different small letters in the Table. Apparently, they are missing.

Response 8: We are very sorry for our mistake to do not double check. We corrected all the figures and tables including footnotes. Thanks

Point 9: L327-329. Figure 1. No subgraphs are observed in Fig. 1.

Response 9: We inserted (A) and (B) at figure 1.

Point 10: Again, there is confusion regarding the Figure they are referring to.

Point 11: There is another confusion regarding Figure.

Response 10 and 11: Again, we are very sorry. We double checked all the figures and Tables in addition to text. Thank you.

Point 12: Apparently, there is confusion on the type of fermentation, although volatile are also produced.

Response 12: We corrected that sentence to make sense.

Point 13: Please, refer to the appropriate

Response 13: We corrected it.

Point 14: Figure 3 should be modified since the legend says “Fermentation time (day) but the intervals are named as fBRE. Please remove fBRe and explain that the graph is referred to fBRE in the legend of the figure.

Response 14: As your comment, we removed fBRE and corrected several mistakes.

Point 15: L403-413. Explanation of EPS production should be clarified.

Response 15: We changed Fig. 1 to Fig. S1 to clarify the explanation on EPS production related with cell biomass and sugar concentration.

Point 16: L449-450. The sentence should be clarified. Is PA-M76 able to synthesize phenols?

Response 16: Of course not. As mentioned in the text, mulberry juice fermented with L. plantarum, L. paracasei, and L. acidophilus showed an increase in the content of phenolic acids, total anthocyanins, and flavanols [32]. Thus, we slightly changed the sentence like: The strain used in this study, PA-M76, might not be sufficient for enhancing the content of phenolic compounds during lactic acid fermentation of BRE, even though TP increased slightly.

Point 17: There is confusion regarding the number of the table.

Response 17: Sorry again. We double checked again and corrected it.

Point 18: L486-518. Were considered for the comparisons the possible different volumes of the volatile present in the different samples analyzed.

Response 18: Yes, definitely.

Point 19: L512 expressions such as “gradually decrease”, “increased” are not fully supported by the experimental results. Please accommodate the expressions to actual data.

Response 19: We deleted “gradually” and inserted actual data.

Point 20: The total variance was wrong.

Response 20: We checked and corrected. Thanks

Point 21: L522. The use of NfBRE and fBRE0 can introduce confusion in the manuscript understanding.

Response 20: Yes, it might be.  However, fBRE0 showed quite different composition with NfBRE when BRE was inoculated by PA-M76 for BRE fermentation due to high inoculation volume as mentioned above. Thus we made list of abbreviation to prevent confusing in the manuscript understanding as shown in the revised manuscript. 

Reviewer 2 Report

General assessment

The study by Song and co-workers focuses on the probiotic properties of a new Pediococcus isolate M76 with emphasis on the formation of an exopolysaccharide. Using a range of analytical methods, it is shown that a black raspberry juice fermented with M76 exhibits such desired beneficial characteristics, such as increased antioxidant and immune stimulatory effects. The experiments are well designed, executed and the results are promising. The manuscript is fluently written and sufficient literature citations are used to support the data well. Overall, the study is certainly of interest to a wider audience and can be published after making a few corrections.

Specific comments

The authors often use abbreviations in the text without further explanation that are not readily apparent to the reader: BRE, EPS GIT, PCA, HCA etc. I strongly recommend a small list of abbreviations.

Line

50: Rubus coreanus in italics

70: a size of 67 kDa seems rather small for EPS; in the literature sizes 500 - 2000 KDa are given for EPS of Pediococcus species. (e.g. Maria Dimopoulou, Aline Lonvaud-Funel, and Marguerite Dols-Lafargue, Chapter 12, Polysaccharide Production by Grapes Must and Wine Microorganisms, Springer International Publishing AG 2017, H. Konig et al. (eds.), Biology of Microorganisms on Grapes, in Must and in Wine, DOI 10.1007/978-3-319-60021-5_12) The authors should comment on this.

84: I assume simulating the GIT

121: Spell out C. elegans

134: LGG?

240: 4.7 ml

268: Mass Hunter database: add literature!

305: Pediococcus acidilactici in italics

Table 1: The items "Temperature", "NaCl" and "Glucose" were not explained in the method section and it is unclear what is meant here in the table. The items "Exopolysaccharides" and "Homology" are superfluous and should be deleted.

314: L. rhamnosus in italics

316: Table 3 heading "Phenolic compounds" in more detail.

318: Table 4 heading a little more precise; what is the control and what is the sample?

339: "previous study". Unattractive repetition

349: Figure 1: Please insert A and B in the graph!

428: Insert A and B in the signature; what conditions were met for B?

439: p<0.05?

457: p<0.05?

461: p<0.05?

References: Check carefully for correct journal style: all species names in italics! Only small letters in the titles! The authors use small and capital letters arbitrarily.

Reference 11 seems incomplete!

Author Response

To reviewer

Thanks for your valuable comments and detail corrections on our manuscript. We revised our manuscript according to your comment as below. Concerning on your specific comments on a small list of abbreviations, I prepared list of abbreviations as you can see in the text. Thanks.

Point 1: L50, Rubus coreanus in italics

Response 1: We corrected it. Thanks.

Point 2: L70, a size of 67 kDa seems rather small for EPS; in the literature sizes 500 - 2000 KDa are given for EPS of Pediococcus species. (e.g. Maria Dimopoulou, Aline Lonvaud-Funel, and Marguerite Dols-Lafargue, Chapter 12, Polysaccharide Production by Grapes Must and Wine Microorganisms, Springer International Publishing AG 2017, H. Konig et al. (eds.), Biology of Microorganisms on Grapes, in Must and in Wine, DOI 10.1007/978-3-319-60021-5_12) The authors should comment on this.

Response 2: Exactly. We changed the sentence and inserted reference as indicated above.

Point 3: L84, I assume simulating the GIT

Response 3: Thanks. We changed all GI track to GIT. (Line 46, 85, 328,339, 347)

Point 4: L121, Spell out C. elegans

Response 4: We corrected it.

Point 5: L134, LGG?

Response 5: The LGG is abbreviation of L. rhamnosus GG (ATCC 53103, LGG) which written in line 82. 

Point 6: L240, 4.7 ml

Response 6: Thanks. We corrected it.

Point 7: L268, Mass Hunter database: add literature!

Response 7: It is a software purchased from Agilent technology. So we added it on text.

Point 8: L305, Pediococcus acidilactici in italics

Response 8: We corrected it. Thanks.

Point 9: Table 1: The items "Temperature", "NaCl" and "Glucose" were not explained in the method section and it is unclear what is meant here in the table. The items "Exopolysaccharides" and "Homology" are superfluous and should be deleted.

Response 9: We removed superfluous items as your comments.

Point 10: L314, L. rhamnosus in italics

Response 10: We corrected it. Thanks.

Point 11: L316, Table 3 heading "Phenolic compounds" in more detail.

Response 11: We changed the heading of Table 3 as “Phenolic compounds of fermented black raspberry extract (fBRE).”

Point 12: L318, Table 4 heading a little more precise; what is the control and what is the sample?

Response 12: We changed “control” to “ Non fermented black raspberry extract (day)” and “sample” to “ Fermented black raspberry extract (day)”

Point 13: L339, "previous study". Unattractive repetition

Response 13: We removed it.

Point 14: L349, Figure 1: Please insert A and B in the graph!

Response 14: We are very sorry. We inserted (A) and (B) and double checked.

Point 15: L428, Insert A and B in the signature; what conditions were met for B?

Response 15: We inserted (A) and (B). The condition was described at line 429.

Point 16: L439, p<0.05?

Point 17: L457: p<0.05?

Point 18: L461: p<0.05?

Response 16-18: Sorry again. We corrected all. Thanks.

Point 19: References, Check carefully for correct journal style: all species names in italics! Only small letters in the titles! The authors use small and capital letters arbitrarily.

Response 19: We checked all the references and corrected them as you indicated. Thanks.

Point 20: Reference 11 seems incomplete!

Response 20: It is not incomplete. We checked it. Thanks!

Round 2

Reviewer 1 Report

Points 1 and 16. Since it is agreed that polyphenols are not synthesized during fermentation, it would be convenient to introduce some explanations on the changes in the polyphenol's contents and supporting them with references. This is particularly interesting in the text.

Author Response

To reviewer

Thanks for your valuable comments on our the revised manuscript. We corrected our manuscript according to your comment as below.

Points 1 and 16. Since it is agreed that polyphenols are not synthesized during fermentation, it would be convenient to introduce some explanations on the changes in the polyphenol's contents and supporting them with references. This is particularly interesting in the text.

Response 1 and 16: L456-461, We added explanations on the changes in the polyphenol's contents with new reference.